# The UK Government's Covid-19 Response and Article 2 of the ECHR (Title I Dignity; Right to Life, Charter of Fundamental Rights of the EU)

**Miroslav Baros**

Faculty of Social Sciences and Humanities, Department of Law and Criminology, Sheffield Hallam University, Sheffield S10 2BQ, UK; m.baros@shu.ac.uk

**Abstract:** The purpose of this article is to assess the impact of the UK government's response to the Covid-19 outbreak from a human rights perspective, particularly its apparent tension with Article 2 of the European Convention on Human Rights (ECHR) in relation to non-Covid-19 patients whose lives were put at risk by not being able to attend appointments and treatments for pre-existing conditions and illnesses. The UK has also rejected the application of the Charter of Fundamental Rights of the European Union with the European Union Withdrawal Act 2018, which will leave the population even more exposed to potential human rights violations. This seems to be a direct consequence of the narrative and slogan employed by the government: "Stay Home; Protect the NHS; Save Lives". Other potentially threatened categories, the NHS staff and prisoners are also mentioned in the same context. The latter have already launched a judicial review application along the same lines: Article 2 of the ECHR and the due regard duty stemming from the Equality Act 2010. The NHS staff were directly at risk, and evidence was emerging almost on a daily basis that implied authorities' responsibility for the shortage of personal protective equipment and testing kits. While there have been a number of discussions on other issues in relation to the lockdown and the strategy directly or indirectly impacting human rights, it appears that no discussion on the impact of the strategy for non-Covid-19 patients and other categories from a human rights perspective has taken place. This gap in analyses and literature merits the present analysis.

**Keywords:** Article 2 of the ECHR; right to life; Charter of Fundamental Rights of the EU; positive duty to protect life

## 1. Introduction

> *"The state finds its highest expression in protecting rights, and therefore should be grateful to the citizen who, in demanding justice, gives it the opportunity to defend justice, which after all is the basic raison d'etre of the State"* (Calamandrei 1992)

The outbreak of Covid-19 at the beginning of 2020 has resulted in an unprecedented response by governments throughout the world that affected population at large at a scale unknown in recent history. Indicative of the prediction expressed in this article is a successful challenge of legality of lockdown in New Zealand. The focus of this article is on the compatibility of the UK government's response, primarily the lockdown and its consequential developments with the right to life under Article 2 of the European Convention on Human Rights (ECHR).

Whilst there have been a wealth of analyses and discussions about the response from human rights perspectives (freedom of expression, freedom of association, the right to protest, the right to liberty and security of person, the right to fair trial, and the right to respect family life and privacy (Council of Europe 2020; Amnesty International UK 2020; Equality and Human Rights Commission 2020;

Maini-Thompson 2020; United Nations Office of the High Commissioner 2020) there is nothing yet about the impact of the lockdown on the right to life of particularly exposed groups that continue to suffer as a result of what the lockdown and its consequences have caused and continue to cause even more than four months since its introduction. The purpose of this article is to examine the impact of the lockdown and the related measures on non-Covid-19 patients, the healthcare workers, and prisoners as well as an increasing tension between the government's response and Article 2 of the ECHR in that respect.

On 24 March 2020, the UK government introduced lockdown measures in an effort to slow the rate of infection of Covid-19 in the country, shielding the National Health System (NHS) and its expected capacity to respond. A slogan was adopted: "Stay Home; Protect the NHS; Save Lives". Arguably, the strategy and measure were accompanied by a strong psychological element and effect on the population through persistent and frequent repetition of the slogan by the Prime Minister himself, as well as by other members of the government. In fact, the government launched an advertisement, which, in addition to the slogan stated above, added: "If you go out, you can spread it. People will die." For the purpose of this article, a question may be posed as to what the individuals can possibly do to *protect* the system that is established to protect themselves, especially those with terminal illnesses, cancer sufferers, patients with cardiovascular diseases, stroke sufferers, etc.

## 2. Chronology

On 28 March, the Prime Minister sent a letter to 30 million households warning that "things will get worse before they get better." Then, he repeated the main reason for the lockdown: " . . . The action we have taken is absolutely necessary, for one very simple reason. If too many people become seriously unwell at one time, the NHS will be unable to cope." (PM Letter to Nation on Coronavirus 2020) It became therefore, quite clear that the only reason for the lockdown was to "protect the NHS" ("to put it simply, if too many people become seriously unwell at one time, the NHS will be unable to handle it—meaning more people are likely to die, not just from coronavirus, but from other illnesses as well" (Nsubuga 2020).

On 26 March 2020, Professor Azra Ghani, report author from the Centre for Global Infectious Disease Analysis (GIDA) said:

> "Acting early has the potential to reduce mortality by as much as 95 per cent, saving 38.7 million lives. At the same time, consideration needs to be given *to the broader impact of all measures that are put in place to ensure that those that are most vulnerable are protected from the wider health, social, and economic impacts of such action*" (emphasis added).

This article relates primarily to the second part of the statement above. The measures, regardless of the rationale and urgency for their introduction, would cause an impact on other groups, on groups other than Covid-19 patients, the NHS staff, prisoners, etc.

It is beyond the merits of this article to discuss the effectiveness of the measures for two reasons: I am not a medical expert, and, secondly, evidence of their effectiveness is not yet available. However, the lockdown measures are directly related to the main point and argument in this article, so I will have to say a few words about their origin and effectiveness.

## 3. "Almighty" Lockdown!

The very origin of the "lockdown" expression and, especially, "quarantine" could be traced back to the 14th century during the spread of plague, when 25 million people, or 60% of Europe's entire population, died (Benedictow 2005) and about a half of the population of England lost their lives. People becoming ill from an infectious fever caused by the bacterium *Yersinia pestis*—likely transmitted from rodents to humans through bites by infected fleas rather than through a virus were dying within hours of the infection, with 80% of the cases ending in mortality. (Augustyn) In a desperate attempt to stop or to slow down the spread, the first lockdown measures were introduced in Europe. *Quarantine*,

on the other hand, originated in Venice at the same time, when the city desperately tried to stop the disease from spreading. Ships arriving in Venice from infected ports were required to sit at anchor for 40 days before landing. This practice was called quarantine, derived from the Italian words *quaranta giorni*, meaning "40 days" (Center for Disease Control and Prevention 2020). Unfortunately, the measure did not stop or slow down the spread. More than 100,000 Venetians died during the outbreak of the plague in the 14th century (Carr 2020).

What transpires from the above is that the very origin of lockdown was pragmatism and epidemiology rather than an effective medical strategy or advice, and this distinction is crucial to my point here, which is that the way the lockdown measures were pursued in the UK had an unintended and unfortunate consequence: By sticking strictly to the narrative of "Stay Home, Protect the NHS", many non-Covid-19 patients' lives were unnecessarily put at risk. (Chakelian and Goodier 2020). On the other hand, the lockdown itself will be instrumental in generating human rights issues because it will affect peoples' immunity, negatively leading to excessive deaths, which in turn, may be put into the context of Article 2 of the ECHR. There are, unfortunately, other categories of individuals whose lives were also put at risk as a result of this narrative, which I will also briefly mention below.

## 4. Article 2 of the European Convention on Human Rights

Under Article 2 of the European Convention on Human Rights (ECHR) "everyone's right to life shall be protected by law". This duty is *positive* in character, and it refers not only to the individual, but to a wider public. According to the guidance of the European Court of Human Rights (ECtHR) on Article 2 of the ECHR, the States' parties to the Convention are not only required to refrain from the intentional and unlawful taking of life, but also to take appropriate steps to safeguard the lives of those within their jurisdiction (The European Court of Human Rights 2020). The point was powerfully made in *Centre for Legal Resources on Behalf of Valentin Câmpeanu v. Romania* para. 130). In broad terms, this positive obligation has two aspects: (a) The duty to provide a regulatory framework and (b) the obligation to take preventive operational measures. In *Mehmet Şentürk and Bekir Şentürk v. Turkey* the Court reiterated that "the positive obligations imposed on the State by Article 2 of the Convention imply that a regulatory structure be set up, requiring that hospitals, be they private or public, take appropriate steps to ensure that patients' lives are protected."

This positive duty requirement has long been confirmed in a number of cases (*L.C.B. v. the United Kingdom*; *Calvelli and Ciglio v. Italy*). The rationale of the duty is to be found in a higher and more permanent standard and appeal to law transcending national borders and requiring state authorities to be more proactive in protecting fundamental rights and dignity, as both the ECHR and the Charter of Fundamental Rights of the EU (CFR) (Bolado 2020) require in the spirit, not only the letter, of the treaties (Dupré 2014). It requires national authorities to act and take necessary measures to protect the rights of the individual. A positive duty is, therefore, the most distinctive and quintessential characteristic of a human rights argument. It transforms freedoms into rights as the ultimate and the most assertive positive entitlements, which give powers to the individual to seek judicial protection by referring to a failure of national authorities to act in a particular set of circumstances.

In the context of healthcare, the positive obligations require States to make regulations compelling hospitals, whether private or public, to adopt appropriate measures for the protection of patients' lives (*Calvelli and Ciglio v. Italy* para. 49; *Vo v. France* para. 89; *Lopes de Sousa Fernandes v. Portugal)*.

In *Aydoğdu v. Turkey* the Court considered that the authorities responsible for healthcare must have been aware at the time of the events that there was a real risk to the lives of multiple patients, owing to a chronic state of affairs that was common knowledge, and yet had failed to take any of the steps that could reasonably have been expected of them to avert that risk. The Court noted that the Government had not explained why taking such steps would have constituted an impossible or disproportionate burden for them, bearing in mind the operational choices that needed to be made in terms of priorities and resources (para. 87). It therefore held that Turkey had not taken sufficient care to ensure the proper organisation and functioning of the public hospital service in this region of the

country, particularly because of the lack of a regulatory framework laying down rules for hospitals to ensure protection of the lives of premature babies in that case.

In addition, an issue may arise under Article 2, where it is shown that the authorities of a Contracting State have put an individual's life at risk through the denial of the healthcare which they have undertaken to make available to the population in general (*Cyprus v. Turkey*, para. 219; *Hristozov and Others v. Bulgaria*, para. 106).

The Court has also accepted that the responsibility of the State, under the substantive limb of Article 2, was engaged as regards the acts and omissions of healthcare providers—firstly, where an individual patient's life was knowingly put in danger by a denial of access to life-saving emergency treatment (*Mehmet Şentürk and Bekir Şentürk v. Turkey)*.

## 5. The Government's Strategy and the Consequences

With all this in mind, I assess the impact of the government's strategy on non-Covid-19 patients, the NHS staff, and prisoners.

First of all, there was an alarming lowering of the rate of admissions to hospitals since the announcement of the measures on 28 March 2020 (Chakelian and Goodier 2020). On 13 April 2020, the official figures indicated that 40.9 per cent of NHS's general acute beds were unoccupied as of the weekend—37,500 of the total 91,600 relevant beds recorded in the data. According to the source, that was 4500 more than the 33,000 the NHS said had been freed up on 27 March, and nearly four times the normal number of free acute beds at this time of year. (West 2020)

Apparently, the reason behind this phenomenon was the slogan "Protect the NHS" because, according to the source above, "The clearout follows a huge ramping up of discharges from hospitals in recent weeks in preparation for the Covid-19 surge, with funding rules and checks scrapped . . . and staff told to focus on discharge, change their thresholds, and be more directive about patients leaving hospitals. The number of patients who have spent 21 days or more in hospital—so-called 'super stranded patients'—has reduced by 40 per cent (West 2020), which will surely lead to harm being done to those who fail to get treatment and widespread suspensions of planned operations.

This is precisely what engages Article 2 of the ECHR and its implicit positive duty to protect life. In fact, in the present context, the UK government *actively* contributed to a failure to protect life of non-Covid-19 patients by encouraging, or, more precisely, by pressuring hospitals to slow down admissions for non-Covid-19 patients, which clearly violates the spirit of the positive duty under Article 2 of the ECHR in the cases stated above, especially in *Calvelli and Ciglio v. Italy* para. 49, where the ECtHR stated: "Those principles apply in the public-health sphere too. The aforementioned positive obligations therefore require States to make regulations compelling hospitals, whether public or private, to adopt appropriate measures for the protection of their patients' lives." Pressuring or incentivising hospitals to speed up discharge from hospitals and to slow down new admissions of non-Covid-19 patients appears to be a very contradiction of the duty referred to by the Court.

## 6. The Slogan and Cancer Patients, Prisoners, and the NHS Staff

There has been a worrying tendency among cancer patients to refuse cancer treatment in hospitals due to fear of catching the virus (Joshi 2020). I will provide just few illustrations and evidence here. According to Professor Charles Swanton, Cancer Research UK's chief clinician, "Cancer survival rates would drop, and the delay in diagnosis and treatment would render some cancers 'inoperable' when they would have been curable if caught earlier." (Wheeler 2020). Then, on 20 April 2020, oncologists expressed their concerns about the impact of the strategy (especially the "Protect the NHS" part) in a letter to the government. They claimed: "In our view, there is a real risk that patients who need proton beam therapy will be denied that treatment and given sub-optimal conventional treatment, which was not theirs nor their clinician's first choice, thus potentially increasing unwanted late toxicities and affecting their quality of life in the long term. Furthermore, parents of children with cancer requiring proton therapy will hope that the Covid-19 situation should not compromise the long-term cure and

quality of life for their children." (Elsom 2020). According to a report by the Institute for Public Policy Research and Carnall Farrar from August 2020 there was a 43% drop in urgent cancer referrals in comparison to the same period last year. (Gregory 2020)

I finally wish to buttress my argument here by reference to the Office of National Statistics figures released on 21 April, according to which there has been a significant increase in non-Covid-19 deaths in the UK in relation to the same period last year. There were 18,500 deaths in the week up to 10 April—about 8000 more than is normal at this time of year. However, 6200 of these deaths were linked to coronavirus and Covid-19, which means that there was a significant increase in deaths of non-Covid-19 patients (Triggle 2020).

According to Karol Sikora, the chief medical officer at the Rutherford Cancer Centre: "Every day, about one thousand new cases of cancer are diagnosed in the UK. This means that, in the seven weeks since the UK shut down to contain coronavirus, roughly 50,000 people should have found out they had cancer. Instead, oncologists and pathology labs across the country have only caught about 10% of those cases."

However, it is not just about cancer patients. "Cardiologists and other healthcare professionals are also sounding the alarm. People have been told to stay at home, so they are not coming into hospitals for checkups or visiting emergency rooms if they feel ill. In April, doctors postponed more than 2 million surgeries to free up 12,000 beds for coronavirus patients, at a potential cost of £3 billion ($3.7 billion)". (Timsit 2020) According to the NHS leaders, "The waiting list for hospital treatment could soar to almost 10 million people by Christmas amid a huge backlog caused by coronavirus disrupting services". (Campbell 2020)

The right to life claim can potentially be made by the NHS staff themselves in the light of increasing credible evidence of failure to provide personal protective equipment (Blackall 2020) to frontline medical staff (Horton 2020). If armed forces can bring Article 2 claims against the government (especially paragraphs 75–86), then surely the NHS staff can do the same. In *Smith and Ors v. Ministry of Defence*, the failure by the Ministry of Defence (MoD) to provide equipment and technology to protect against the risk of friendly fire fell within the MoD's duty of care on the grounds that it would be fair, just, or reasonable to extend the duty (para. 101) (UK Parliament Defence Committee 2014).

Finally, prisoners are also impacted by the strategy, and a judicial review application has already been launched. According to the Pre-Action protocol letter sent to the Lord Chancellor and the Secretary of State for Justice by Bhatt Murphy Solicitors on 17 April 2020: "The rate of infection following tests is increasing rapidly. Since our clients first wrote to you jointly on 27 March 2020, the number of prisoners infected has increased from 27 on 26 March 2020 to 232 as of 15 April 2020, an increase of almost ten-fold." It was clearly only a matter of time before some of these issues started to come before the European Court of Human Rights. Expectedly, at the beginning of July 2020, a case involving the UK Government concerning the impact of Covid-19 on conditions of detention in prison was communicated (*Hafeez v UK*).

The Pre-Action letter mentioned above also reminds the government of the positive duty, which is the crux of my argument here: The duty applies to all individuals who are detained (*Keenan v. United Kingdom* para. 111; *Kudla v. Poland* para. 94). *Keenan* established the uncontroversial proposition that one of the reasons the state owes this duty is because of the inherent vulnerability of those who are detained by the state, para. 110. The duty is "particularly stringent in relation to those who are especially vulnerable by reason of their physical or mental condition" (*Rabone v. Pennine Care NHS Foundation Trust* para. 22 per Lord Dyson (Howard League 2020).

## 7. Conclusions

*"Progress is not an illusion, it happens, but it is slow and invariably disappointing . . . Consequently two viewpoints are always tenable. The one, how can you improve human nature until you have changed the system? The other, what is the use of changing the system before you have improved human nature?"* (Orwell 1969)

As suggested in this article, the duty to protect life under Article 2 of the ECHR requires positive action, meaning not only pre-emptive, but also proactive, forward-looking and lateral thinking as to what needs to be done to comply with the Article and the spirit of the ECHR and CFR. It unfortunately transpires that the strategy (by which I mean an inevitable psychological element, which was seemingly given almost in a state of panic by the government) of "Protect the NHS" had created conditions that are already proving harmful for non-Covid-19 patients and other categories as identified in this article. Using powerful slogans by the government in order to cause an exaggerated sense of fear of the virus among the population, thereby actively discouraging people from seeking medical assistance for conditions unrelated to Covid-19 seems to be contrary to the very spirit of both the ECHR and CFR and the positive duty to protect life. There are also a number of other aspects indicating a tension between the strategy and human rights (Arts. 3, 8, 9, 10, 14). (Hoar 2020)

One has to admit, though, that the government seemed to be aware of the problem, and, since the middle of April 2020, the ministers were pleading with non-Covid-19 patients to make appointments and go for treatments. However, this plea can also be seen as a recognition of creating the problem in the first place.

This episode demonstrates how fragile and uncertain the status of fundamental values may be, even today, when social, democratic, and economic progress made us all take them for granted. The CFR (rejected by the UK with the European Union Withdrawal Act 2018) (Blackstone Chambers 2018) by placing the right to life within "dignity", implicitly elevating the fundamental character of the right, provided an opportunity for the member states to adopt a more protective stance towards the right to life. It seems, therefore, that rejecting the Charter and the European Court of Justice (ECJ) robust jurisdiction in matters such as those presented in this article will clearly have a huge impact on the modern concept of protection and promotion of human rights in the UK.

**Funding:** This research received no external funding.

**Conflicts of Interest:** The author declares no conflict of interest.

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
