# Peer review of "The UK Government’s Covid-19 Response and Article 2 of the ECHR (Title I Dignity; Right to Life, Charter of Fundamental Rights of the EU)"

_laws_

Round 1
Reviewer 1 Report
This is a well evidenced piece on an (as yet) under-researched area.
The writing might be tightened up very slightly in places, but the style is a nice combination of doctrinal analysis and more anecdotal discussion.
Author Response
Many thanks for your review. English tightened up and improved now.
Reviewer 2 Report
Reviewer response:
Thank you for the opportunity to review the manuscript "UK Government's Covid-19 response and Article 2 ECHR"
The Introduction does not clearly set out the purpose of the article, why the article is significant/important (i.e. the gap in the literature), nor does it introduce the flow of the manuscript’s narrative to the reader.
This manuscript needs a good proof read.
Abstract
- Line 10: why “(here)” – usually Abstracts do not include references, let alone weblinks.
Introduction
- Line 28: Use of two colons?
- First para can be tightened, content is repetitive. Need to also insert references to support statements of fact.
- Lines 32-33: “By the end of the very first day of the campaign it started reverberating in my mind:…” Lines 35-36: “I did scratch my head though in relation to one of those in particular…” These are not sentences indicative of a strong journal article to follow… (not boding well given this is the first paragraph)
Author Response
Many thanks for your review. English tightened up and improved now. I revised the article from that particular perspective and I hope it looks better now to a native English speaker. I have also addressed another comment explicitly, which was that I have not stated the purpose of the article (not correct) by including a separate passage on the purpose of the article.